# The Complex Interactions Between HIV-1 and Human Host Cell Genome: From Molecular Mechanisms to Clinical Practice

**DOI:** 10.3390/ijms26073184

**Published:** 2025-03-29

**Authors:** Manlio Tolomeo, Francesco Tolomeo, Antonio Cascio

**Affiliations:** 1Department of Health Promotion Sciences, Maternal and Infant Care, Internal Medicine and Medical Specialties, University of Palermo, 90127 Palermo, Italy; antonio.cascio03@unipa.it; 2Department of Infectious Diseases, (Azienda Ospedaliera Universitaria Policlinico) A.O.U.P. Palermo, 90127 Palermo, Italy; 3Department of Biological, Chemical, and Pharmaceutical Sciences and Technologies (STEBICEF), University of Palermo, 90127 Palermo, Italy; francesco.tolomeo01@unipa.it

**Keywords:** HIV-1, virus–host interaction, viral DNA integration, DNA damage, latency, LEDGF/p75, capsid, CPSF6, HIV-1 therapies

## Abstract

Antiretroviral therapy (ART) has significantly improved the prognosis of human immunodeficiency virus type 1 (HIV-1) infection. Although ART can suppress plasma viremia below detectable levels, it cannot eradicate the HIV-1 DNA (provirus) integrated into the host cell genome. This integration often results in unrepaired DNA damage due to the HIV-1-induced inhibition of DNA repair pathways. Furthermore, HIV-1 infection causes telomere attrition in host chromosomes, a critical factor contributing to CD4+ T cell senescence and apoptosis. HIV-1 proteins can induce DNA damage, block DNA replication, and activate DNA damage responses across various organs. In this review, we explore multiple aspects of the intricate interactions between HIV-1 and the host genome involved in CD4+ T cell depletion, inflammaging, the clonal expansion of infected cells in long-term-treated patients, and viral latency. We discuss the molecular mechanisms of DNA damage that contribute to comorbidities in HIV-1-infected individuals and highlight emerging therapeutic strategies targeting the integrated HIV-1 provirus.

## 1. Introduction

Human immunodeficiency virus type 1 (HIV-1) is an RNA virus that belongs to the retrovirus family. Retroviruses are unique among virus families due to their ability to integrate their genetic material into the DNA of host cells. This integration occurs after the production of double-stranded linear viral DNA (vDNA) from viral RNA, a process known as reverse transcription, which is catalyzed by the viral enzyme reverse transcriptase (RT) [1,2,3]. The integration of linear vDNA marks a “point of no return” for the infected cell, making it a permanent carrier of the viral genome, referred to as a provirus.

Retrovirus infections in mammals over millions of years have introduced proviral sequences into their genomes. These sequences have been transmitted through retroviral infections of the germline. Approximately 8% of the human genome consists of sequences derived from retroviruses, predominantly non-coding regulatory DNA often maintained in a transcriptionally silent state by the host cell’s repressive DNA methylation machinery [4]. Mammals have co-opted some endogenous retroviruses for physiological functions; however, these sequences can occasionally have adverse health effects [5]. In contrast to human endogenous retroviruses, HIV-1 proviral integration or components of HIV-1, such as regulatory or accessory proteins, can interact with the human genome, leading to DNA damage or gene dysregulation. Integration can result in DNA breaks [6,7,8], which are often left unrepaired because HIV-1 can inhibit the host DNA repair machinery [9,10]. Proviral integration within or close to specific genes—such as oncogenes and KRAB-ZNF genes—can confer a proliferative advantage, resulting in the clonal expansion of latently infected cells [11,12]. Furthermore, active cellular replication to compensate for the loss of CD4+ T cells, HIV-1-mediated inhibition of DNA repair mechanisms, and the accessory HIV-1 Tat protein can contribute significantly to the shortening of telomeric host DNA, which is a key driver of CD4+ T cell senescence and apoptosis [13,14]. Finally, HIV-1-induced genetic damage in cells across various organs can contribute to the development of comorbidities, including pulmonary arterial hypertension, HIV-1-associated nephropathy, and neurological dysfunctions [15,16,17]. This review examines various aspects of the complex interactions between HIV-1 and the host genome. These interactions are critical in the pathogenesis of CD4+ T cell depletion, inflammation, viral latency, and the clonal expansion of latently infected cells in long-term-treated patients. We also explore novel insights into the role of the viral capsid and its inhibitor, Lenacapavir. Finally, we discuss the molecular mechanisms of DNA damage that contribute to comorbidities in individuals living with HIV-1. We highlight emerging therapeutic strategies targeting the integrated HIV-1 provirus, such as the “Shock and Kill” and the “Block and Lock” approach, as well as CRISPR-based gene editing.

## 2. Integration and Post-Integration Stages in HIV-1 Infection

### 2.1. HIV-1 Provirus Integration

HIV-1 infection begins with the interaction of the viral envelope (Env) glycoproteins with the CD4 receptor and the CCR5/CXCR4 coreceptors expressed on the plasma membrane of the target immune cells. The Env glycoprotein Gp160 is a viral surface protein encoded by the HIV-1 *env* gene. This glycoprotein is cleaved by a host furin-like protease into the receptor-binding fragment gp120 and the fusion fragment gp41. The interaction between gp120 and the CD4 receptor results in a conformational change of gp120 that allows a secondary interaction of gp120 with the cellular coreceptor CCR5 or CXCR4 and the insertion of the N-terminal of gp41 into the cell membrane. Subsequently, gp41 undergoes a conformational change that brings the virus closer to the cell, allowing the viral membrane to fuse with the cell membrane and the viral capsid to enter inside the cell [18]. The reverse transcription follows the capsid entry. The enzyme RT converts the single-stranded RNA genome into an integration-competent double-stranded DNA that is transported into the nucleus and integrated into the host DNA through the viral enzyme integrase [18].

Many concepts about the early stages of HIV-1 infection have changed based on recent studies that have demonstrated the importance of the capsid in regulating reverse transcription, nuclear entry, integration, viral assembly, and maturation [19]. Inside the capsid, the vDNA molecule exists in the form of a preintegration complex (PIC) consisting of the viral proteins Vpr, matrix proteins, integrase, host proteins (Barrier-to-autointegration factor 1), and the vDNA [20]. Matrix proteins and Vpr both contain nuclear localization signals (NLS), so previous studies indicated that the PIC could play a role in nuclear import [21,22,23,24]. However, more recent studies have shown that the capsid is responsible for the nuclear import of the PIC [25,26,27]. The capsid is transported through the cytoplasm towards the nucleus by the motor proteins kinesin and dynein. When the capsid arrives at the nuclear pore complex (a large protein complex of the nuclear pore), it is directed toward the central channel of the nuclear pore via interaction with Nup358 (nucleoporin 358). Subsequently, the capsid interacts with other essential proteins: Nup153 and CPSF6 (Cleavage and polyadenylation specificity factor subunit 6). These proteins allow the translocation of the capsid into the nucleoplasm, where reverse transcription is completed. CPSF6 not only functions in nuclear import but also directs the viral complex enclosed in the capsid to the nuclear speckles, where integration of the viral DNA into the host cell genome occurs [28].

### 2.2. Post-Integrational DNA Repair

The integration of retroviral DNA by viral integrase creates a discontinuity in the chromatin of the host cell, and repair of this damage is necessary to complete the integration process. Before integration, viral integrase hydrolyzes a phosphodiester bond at each vDNA end, excising a dinucleotide and exposing 3′-OH groups on cytosine–adenine dinucleotides. The 3′-vDNA ends are then inserted into the host genome, creating five-nucleotide gaps flanking the integrated vDNA [29]. The host DNA repair machinery subsequently fills these gaps and removes the remaining dinucleotides at the 5′ ends of the integrated vDNA [29,30,31,32]. Several studies have demonstrated the involvement of DNA-dependent protein kinase (DNA-PK) in HIV-1 post-integration DNA repair [30,33,34]. Given that DNA-PK primarily functions as a sensor in DNA double-strand break (DSB) repair, its role in HIV-1 post-integration repair was initially considered unexpected, as the HIV-1 integration intermediate does not contain DSBs. This involvement was clarified when the formation of the Ku70–integrase complex was observed during the DNA damage response (DDR) in HIV-1 post-integration repair [30,34]. Ku is a dimeric protein complex composed of two polypeptides, Ku70 and Ku80, that binds to DNA DSB ends and is required for the non-homologous end joining (NHEJ) pathway of DNA repair [35]. In HIV-1-infected cells, the viral integrase recruits Ku70 at the site of integration. This polypeptide then attracts Ku80, DNA-PK, and ATM (ataxia-telangiectasia mutated). ATM is a serine/threonine protein kinase that phosphorylates CHK2 (checkpoint kinase 2) and H2AX, two key proteins that initiate activation of the DNA damage checkpoint (Figure 1). The serine/threonine protein kinase CHK2 inhibits CDC25, preventing the entry of the cell into the S and M phases. H2AX is a histone protein from the H2A family that contributes to nucleosome formation, chromatin remodeling, and DNA repair. The phosphorylated form of H2AX is a scaffold that attracts repair factors to DNA damage sites and retains them until the damage is removed. As we will report in the following sections, components of HIV-1 can inhibit ATM-directed DNA repair processes and CHK2, causing DNA repair failure and leading to the dysregulation and apoptosis of infected cells.

### 2.3. HIV-1 Provirus Integration Sites

HIV-1 provirus integration is not entirely random but is influenced by a preference for more accessible areas of the genome, often related to a more open and active chromatin structure and transcriptional factors that facilitate viral replication (Figure 2) [36,37,38,39,40]. The number of integration sites varies depending on several factors, such as the class of patient (treated versus untreated, acute versus chronic infection, or untreated aviremic elite controller versus treated aviremic patients), the type of cells examined (activated or resting), the kind of treatment, and the experimental techniques used. No univocal data are available on the exact number of integration sites [41,42,43,44].

Integration has been observed in intronic and exonic regions of active genes. These regions are prevalently involved because they contain the necessary machinery for transcription and splicing, which HIV-1 may exploit for replication. HIV-1 provirus can also integrate into intergenic regions between genes that are generally not transcriptionally active and contain regulatory elements [45,46].

The actively transcribed genes in which HIV-1 preferentially integrates are often located in speckle-associated and speckle-proximal chromatin subnuclear domains (Figure 2). These are regions of chromatin organized near nuclear speckles, which are membrane-less nuclear structures enriched with RNA splicing factors and transcription-related proteins. Genes located into or near nuclear speckles are often transcribed and processed more efficiently, as the high concentrations of transcription and RNA processing factors are more readily accessible to these genes [47]. These regions of chromosomes usually show epigenetic modifications of histones that favor integration [48,49]. The histone epigenetic modifications observed in HIV-1 integration sites are the following: (i) histone H3 lysine 4 trimethylation (H3K4me3) is the mark of actively transcribed gene promoters and enhancers and is the site of RNA Polymerase II recruitment; (ii) histone H3 lysine 9 acetylation (H3K9ac) and H3 lysine 27 acetylation (H3K27ac), found in active chromatin and enhancer regions, are histone epigenetic modifications associated with open euchromatin that facilitates HIV-1 DNA integration; and (iii) histone H3 lysine 36 trimethylation (H3K36me3), found in actively transcribed gene bodies, is an epigenetic modification of histone H3, which is a key protein in DNA packaging. SETD2 histone methyltransferase catalyzes the trimethylation of H3K36 in H3K36me3. In vitro integration assays have shown that the trimethylation of lysine 36 of histone H3 significantly increases the integration of HIV-1 PIC into nucleosomal substrates, indicating that this epigenetic modification is significant for the integration of the virus [50,51]. However, SETD2 or H3K36me3 is not strictly required for HIV-1 infection; instead, they influence the location of integration sites and play a key role in regulating HIV-1 post-integration expression and latency [52]. In contrast, latent HIV-1 reservoirs often reside in regions that undergo epigenetic silencing, marked by H3K9me3 and H3K27me3 (repressive marks) [53].

Two host proteins, Pol II (RNA polymerase II) and LEDGF/p75 (Lens Epithelium-Derived Growth Factor), play key roles in integrating HIV-1 provirus into the human genome, directly contributing to the positions and efficiency of integration. Pol II plays an indirect but essential role in integrating the HIV-1 provirus into the host genome. Pol II-associated transcriptional activity creates a favorable environment for integration by loosening chromatin. It generates transcription-associated chromatin marks (e.g., histone modifications such as H3K36me3) that are recognized by the HIV-1 integration machinery, specifically by LEDGF/p75 [54]. LEDGF/p75 binds to HIV-1 integrase and directs the integrase complex to the integration sites, interacting with epigenetically modified histones, especially H3K36me3 [55,56,57,58,59,60].

The analysis of integration sites has shown that HIV-1 integration occurs predominantly in specific genes encoding proteins involved in cell proliferation (such as BACH2) [61,62], cancer (such as MKL2 and HN1L, two genes implicated in sarcomas and non-small cell lung cancer, respectively) [61,63,64,65], chromatin modification (such as KANSL1) [61,66], and pre-mRNA splicing (such as PRPF6) [61,67]. Several studies have reported that the most frequent HIV-1 integration sites are MKL2 and BACH2 [11,68,69,70,71]. In these genes, the HIV-1 provirus is often integrated in the same transcriptional orientation, a condition that can promote the expression of host genes. Beyond its role in sarcomas, MKL2 is also implicated in chondroid lipomas [72] and has been shown to mediate cancerous transformation in DLC1-deficient hepatocellular and mammary carcinoma cells [73]. Although BACH2 promotes the B-cell receptor-induced proliferation of B lymphocytes and represses cyclin-dependent kinase inhibitors [62], it has also been identified as a novel partner gene in IGH translocation in a patient with highly aggressive B-cell lymphoma/leukemia [74]. Furthermore, the insertion of proviruses at the BACH2 locus in a murine model of B-lymphomagenesis provides strong evidence that mutations at this locus contribute to disease development [75].

Other genes that are frequent sites of integration are *ARIH2* (a cancer-related gene involved in acute myeloid leukemia and gastric cancer) [61,76,77], *MKL1* (a gene related to *MKL2*), *MROH1* (encoding large multiple HEAT-repeat-containing proteins) [61,78], and *XPO6* (encoding an importin-beta protein that mediates the nuclear export of profilin–actin complexes) [61,79].

In contrast to actively infected cells, the integration sites of provirus in resting cells are often found in suboptimal regions for proviral gene expression, which seems to promote the latency of HIV-1 [44,68,80,81,82]. Latency is the major obstacle to HIV-1 infection cure. If ART treatment is interrupted, the virus starts replicating from reservoirs of latently infected cells, so ART must be taken for life to suppress viral replication continually. The sites of integration in patients treated over time with ART or in elite controllers (individuals with natural control of HIV-1) have been found in heterochromatin, non-genic regions, and in an opposite orientation to the host gene, suggesting that a selection of intact latent integrated HIV-1 occurs during prolonged antiretroviral therapy or in subjects able to control viremia without ART [83,84,85]. Latency is favored by specific epigenetic modifications of histones, including histone methylation and acetylation. Histone 3 methylation of lysine 27 (H3K27) and histone 3 methylation of lysine 9 (H3K9) are frequently observed in resting cells [86]. These modifications alter chromatin accessibility and have an inhibitory effect on the transcriptional activity of the integrated provirus [86,87].

Histone modifications play a key role in latency; however, several studies suggest that latency primarily depends on a combination of factors. In addition to epigenetic modifications of histones, these include low levels of key activating transcription factors, provirus orientation, and integration into specific genes, such as *KRAB-ZNF* genes [53,88,89,90].

### 2.4. HIV-1 Provirus Orientation

When the provirus integrates into a host gene, its orientation concerning the transcriptional mechanism of the host gene can influence its expression and viral latency (Figure 2). This effect is primarily due to transcriptional interaction and interference between host gene transcription and the integrated provirus. Researchers inserted HIV-1 genomes into the intron of an active host gene in both orientations to assess the impact of host gene readthrough transcription. They found that when the provirus was in the same orientation as the host gene, readthrough transcription enhanced HIV-1 gene expression. Conversely, in the opposite (convergent) orientation, readthrough transcription inhibited HIV-1 expression. This orientation had a more than tenfold effect on HIV-1 gene expression [91].

An analysis of HIV-1 integration sites in primary CD4+ T cells revealed that the orientation of the provirus relative to the host gene’s transcriptional direction was a critical determinant of latency. Proviruses integrated in the same orientation as host genes were transcriptionally active, while those in the opposite orientation tended toward latency [92]. The orientation of provirus can modulate its expression even when integrated in proximity to active transcriptional units. Einkauf et al. conducted a detailed analysis of the chromosomal positions of many intact and defective proviral sequences in individuals on long-term antiretroviral therapy. The results of their study indicate that prolonged antiretroviral treatment is associated with an accumulation of intact proviruses with deeper latency characteristics, probably caused by immune-mediated selection pressure. They observed that the insertion sites were near active transcriptional units and accessible host chromatin sites and that the intact viral sequences were located in opposite orientations to host genes, supporting the assumption that transcriptional interference between host and proviral gene expression may be a predominant mechanism for maintaining HIV-1 latency, and explaining why HIV-1 can remain transcriptionally silent despite integration into actively transcribed and typically highly expressed host genes [83]. The role of ART in selecting cells with latency characteristics and the specific orientation of provirus was studied by Kok et al. [61], who examined the influence of the activation state of CD4^+^ T cells, the effect of antiretroviral therapy, and the clinical stage of HIV-1 infection on HIV-1 integration site features and selection. They observed a preference for a convergent orientation of the provirus compared to the same orientation relative to the host gene in all HIV-1 integration sites. This was more evident in patients treated with ART compared with untreated patients. Although provirus orientation is probably not the only factor determining HIV-1 latency, understanding these events that occur over time during ART could be essential for developing therapies that aim to reverse latency or eliminate reservoirs of HIV-1.

## 3. HIV-1-Induced Host Genome Damage

### 3.1. HIV-1-Induced DNA Breaks

The HIV-1 genome contains nine genes that encode fifteen viral proteins [93]. The mRNAs for viral structural proteins are transcribed by the *gag*, *pol*, and *env* viral genes. In addition, HIV-1 encodes for proteins with essential regulatory functions (Tat and Rev) and for proteins with accessory functions, such as Nef, Vpr, Vif, and Vpu. Accessory HIV-1 proteins can interfere with post-integrational repair and induce further strand breaks in the DNA of infected cells. Viral protein R (Vpr) is expressed in five primate lentiviral groups (HIV-1, HIV-2, and SIVs from mandrill, Sykes, and African green monkeys) [94]. It is a multifunctional protein that plays several functions in the viral life cycle, such as increasing the rate of viral replication and accelerating the cytopathic effect of the virus in T cells and macrophages [95]. Several reports indicate that Vpr induces DDR, suggesting that this viral protein may itself cause damage to host DNA [6,7,8,9]. However, the mechanism by which Vpr induces structural alterations in DNA remains unclear. Several mechanisms of Vpr-induced DNA damage and DDR activation have been proposed. Experiments conducted by Li et al. showed that Vpr directly induces DNA damage, blocks DNA replication, and activates ATR (ataxia telangiectasia and Rad3-related protein) [9]. Iijima et al. [8] demonstrated that Vpr unwinds double-stranded DNA by converting it into a relaxed form and induces the ubiquitination of histone H2B. This chromatin remodeling promotes the efficient loading of RPA70 (replication protein A), a single-stranded DNA-binding subunit of RPA that activates the ATR-dependent DDR [96,97,98]. In addition, the Vpr-induced unwinding of double-stranded DNA resulted in the accumulation of negatively supercoiled DNA and covalent complexes of topoisomerase 1 and DNA, which cause DSBs (Figure 3) [7]. While topoisomerase 1 is responsible for causing single-strand breaks, under certain conditions—such as collisions with the replication machinery—these single-strand breaks can be converted into double-strand breaks, which are much more dangerous and can lead to genomic instability. Another mechanism by which Vpr can induce the formation of DSBs in DNA is through the production of ROS, which leads to oxidative stress. This oxidative damage also interferes with cellular repair mechanisms [7]. By causing replication fork stalling, Vpr indirectly leads to the collapse of these forks, a phenomenon associated with the formation of DSBs. This is often compounded by the inability of the cell to resolve stalled forks due to the Vpr-mediated inhibition of repair proteins [99]. The Vpr-induced DNA damage and activation of DDR signaling can alter cellular transcription through the modulation of NF-kB. This primary transcription factor family regulates the innate and adaptive immune responses and inflammation genes. Following DSBs induced by Vpr, DNA damage activates ATM, which results in NEMO activation and the IKK-mediated degradation of IkB. This causes the translocation of the NF-kB class II type RelA into the nucleus. RelA binds to genes containing NF-kB sites, promoting their transcription. Since the HIV-1 genome contains multiple NF-kB binding sites in its promoter regions, an advantage of Vpr’s activation of NF-kB signaling via ATM is the potential direct enhancement of HIV-1 gene expression. Furthermore, the increased activation of NF-kB leads to the production of pro-inflammatory factors such as IFNα, IFNλ1, IRF1, IRF7, TNFα, IL-6, and IL-8, which play a key role in determining the state of chronic inflammation in HIV-1-infected individuals [100].

Of interest is that Vpr promotes the degradation of several DDR proteins, inhibiting double-strand DNA break repair by recruiting the CRL4A^DCAF1^ ubiquitin ligase complex [9]. CRL4^ADCAF1^ (also known as the CRL4 E3 ubiquitin ligase complex) is a specific variant of the CRL4 (Cullin-RING Ligase 4) complex, a type of E3 ubiquitin ligase. These complexes tag proteins with ubiquitin, signaling their degradation by the proteasome. For example, Vpr-expressing cells cannot repair DSBs induced by etoposide or other DSB-inducing compounds [9]. Li et al. observed that cells transfected with HIV-1 and HIV-2 Vpr decreased DNA repair by homologous recombination by 66% and 49%, respectively [9]. Homologous recombination repairs DNA before the cell enters mitosis (M phase), during the end of the S and G2 phases [101]. Cells often utilize non-homologous end joining (NHEJ) to repair DSBs when homologous recombination is repressed [102]. Similar to homologous recombination, HIV-1 Vpr expression also decreased NHEJ efficiency by 51%. Differently from HIV-1, HIV-2 Vpr did not significantly decrease NHEJ, indicating functional differences between HIV-1 and HIV-2 Vpr [9]. DDR is also inhibited by the HIV-1 accessory protein Vif (virion infectivity factor) [10]. Vif is an HIV-1 accessory protein required to counteract the antiviral activity of host APOBEC3 DNA cytosine deaminases. APOBEC3 (apolipoprotein B mRNA editing enzyme, catalytic subunit 3G) is a family of proteins that can bind to nascent HIV-1 particles and catalyze cytidine deamination to uridine in the single-stranded DNA substrate. This results in nonsense mutations, missense mutations, and abortive integration. The interaction of APOBEC3G with Vif results in the recruitment of APOBEC3 into a complex consisting of ubiquitin ligase E3, scaffold protein cullin5 (CUL5), and the substrate adaptors elongin B (ELOB) and elongin C (ELOC). This promotes the polyubiquitination and degradation of APOBEC3 and inhibits the APOBEC3-mediated antiviral defense [103]. In addition, Vif antagonizes PP2A (protein phosphatase 2A) activity [10]. PP2A is an intracellular protein phosphatase that regulates cell growth and metabolism, DNA replication, DNA damage checkpoints, and DNA repair processes. PP2A governs the activation of kinases involved in DDR, such as ATM, ATR, DNA-PK, CHK1, and CHK2 [104,105,106,107]. PP2A antagonism mediated by Vif causes the accumulation of DSBs in the host DNA through the inactivation of kinases involved in DDR. However, pharmacologic and functional studies suggest that Vif blocks Vpr-directed ATM activation but not ATR [10]. In addition, Vif blocks the ATM-directed induction of NF-kB signaling and phosphorylation [103]. The inhibition of cellular DNA repair mechanisms has essential implications in CD4+ T cell depletion, inflammaging, and several comorbidities observed in HIV-1-infected patients (Figure 2).

### 3.2. DNA Damage and CD4+ T Cell Depletion

HIV-1 causes CD4+ T-cell depletion not only by direct cytopathic effects (which generally occur in the acute phase of infection) but also by direct and indirect mechanisms that damage or alter DNA (Figure 2). Being long-lived, CD4+ T cells are exposed for a long time to genomic insults that may affect different areas of DNA. As previously described, HIV-1 can inhibit several DNA damage repair mechanisms so that infected CD4+ T cells have a limited ability to repair genomic insults. This condition can induce the premature senescence and apoptosis of infected cells [108]. Furthermore, in HIV-1 infection, the replicative pressure to replace dead CD4+ T cells causes progressive telomere shortening (a phenomenon defined as telomere attrition), which is a hallmark of cellular senescence [109,110]. Telomeres are specialized nucleoprotein structures located at the ends of linear chromosomes in eukaryotic cells, serving to protect genetic information from degradation over successive rounds of DNA replication. During DNA replication, the enzyme DNA polymerase encounters a fundamental limitation when replicating the ends of linear DNA, resulting in the progressive shortening of chromosomes with each cell division. This phenomenon, known as the end-replication problem, arises because DNA polymerase can only synthesize DNA in the 5′ to 3′ direction, requiring a primer to initiate synthesis and adding nucleotides to the 3′ end of the growing DNA strand. Consequently, the leading strand is synthesized continuously in the 5′ to 3′ direction, whereas the lagging strand is synthesized discontinuously as Okazaki fragments, each necessitating an RNA primer. At the extreme end of the lagging strand, the final RNA primer provides a starting point for DNA synthesis. However, once this primer is removed, DNA polymerase is unable to replace it with DNA, as there is no upstream 3′-OH group to which nucleotides can be added. Since this missing segment of DNA cannot be replaced, the newly replicated chromosome lacks a small portion at the 3′ end of the lagging strand. Over successive rounds of replication, this leads to the progressive shortening of chromosomes. To mitigate this, eukaryotic chromosomes possess specialized telomeric sequences at their ends—composed of repetitive TTAGGG sequences in humans—which serve to protect essential genetic information from degradation [111]. When telomeres become critically short, cells enter a state called senescence (a non-dividing state) or undergo apoptosis, contributing to aging.

T lymphocytes can undergo a finite number of replications before entering a state of senescence and cell death. This limit to cellular replication depends on the erosion of telomeres that occurs after each cell division. Upon activation, T cells can proliferate by activating the enzyme telomerase, which can add telomeres to the ends of chromosomes, thus preventing the development of replicative senescence [112]. However, after repeated stimulation, highly differentiated T cells lose the ability to induce this enzyme [113]. To understand the role of telomeres in HIV-1 infection, Zhao et al. studied the homeostasis of T cells, telomeric DNA repair, and DNA damage mechanisms in HIV-1-infected individuals on ART with undetectable viremia. They observed that CD4+ T cells of these patients were susceptible to DNA damage that extended to the ends of chromosomes, leading to accelerated telomere erosion, mainly due to a suppression of the expression and activity of ATM and its downstream CHK2. In contrast, the DNA damage sensor MRN complex remained intact. Interestingly, ectopic ATM expression by transfection of HIV-1-derived CD4+ T cells with a Flag-His-ATM construct reduced DNA damage, apoptosis, and cellular dysfunction [13]. Considering that telomere integrity is crucial for cell survival, Khanal et al. conducted a series of in vitro studies to determine whether HIV-1-induced T-cell apoptosis is related to telomere damage [14]. They determined the sites of DNA damage by confocal microscopy by assessing the presence of 53BP1-induced nuclear foci. This analysis is based on the observation that following a genotoxic insult, 53BP1 functions as a docking site for other DNA repair factors to form nuclear foci visible under the microscope [113]. Dysfunctional telomere-induced 53BP1 was higher in primary CD4+ T cells after HIV-1 infection than in uninfected CD4+ T cells. Infected cells showed a marked shortening of telomeres compared to uninfected cells. Of note, the protease inhibitor Raltegravir partially blocked or restored telomere lengths. Finally, they observed that the PI3K/ATM pathway suppression in infected cells correlated with telomeric DNA damage and premature CD4+ T cell aging and apoptosis. Based on these findings, the authors suggested a model of CD4+ T cell depletion in HIV-1 infection in which the suppression of PI3K/ATM promotes telomeric DNA damage and erosion. This results in premature cell aging and apoptosis. ART cannot fully restore the PI3K/ATM pathway in HIV-1-infected cells, so infected cells would live in a state of high apoptotic susceptibility. In a univariate analysis, Alejos et al. observed that a shorter telomere length was associated with older age, HIV-1 RNA ≥ 100,000 copies/mL, CD4 count < 200 cells/μL, a lower CD4:CD8 ratio, statin treatment, and current alcohol consumption [114]. In another study, higher levels of immune activation increased soluble CD14 levels, and a high percentage of CD38(+) HLA-DR(+) cells among CD4+ T cells correlated with shorter telomeres [115].

Among telomere shortening causes, the accessory HIV-1 Tat protein plays a relevant role [116]. Tat protein can decrease the telomerase level in infected cells, and this reduction occurs in uninfected T cells when exposed to Tat protein (Figure 2) [116]. The inhibition induced by the Tat protein appears to be related to the ability of this protein to inhibit the nuclear levels of hTERT (human telomerase reverse transcriptase), which is the catalytic subunit of telomerase with reverse transcriptase activity [116]. In addition, this protein alters the AKT pathway and the molecular interaction with chaperones required for hTERT phosphorylation, nuclear import, and activation [117]. ART initiated in the early stages of HIV-1 infection can slow telomere shortening in CD4+ T cells [117]. In contrast, ART administered at a late stage of infection is associated with significant and sustained telomere length shortening [118]. In HIV-1-infected adults with prolonged virological suppression, NRTI-containing ART was associated with smaller gains in blood telomere length after 2 years of follow-up [119]. Tenofovir (tenofovir disoproxil fumarate and tenofovir alafenamide fumarate) and abacavir are potent inhibitors of human telomerase in vitro, likely due to the homology between HIV-1 RT and hTERT [118,119,120,121,122]. In cross-sectional studies of patients treated with ART, longer exposure to tenofovir was associated with the inhibition of telomerase activity and significantly shorter telomere length in CD4+ T cells [123]. A longitudinal study of long-term virologically suppressed patients showed that treatment with tenofovir or abacavir was associated with a smaller gain in telomere length than regimens with a sparing of NRTIs [119]. Furthermore, an increase in blood telomere length was observed in HIV-1-infected patients who switched to a therapeutic simplification with dolutegravir plus lamivudine compared to patients who continued on a triple regimen containing NRTIs [124]. Finally, a study on European children with HIV-1 acquired in the perinatal period and treated early showed that the size of the HIV-1 reservoir is associated with telomere shortening and immunosenescence [125].

### 3.3. DNA Damage and Its Role in Inflammaging in HIV-1-Infected Individuals

Inflammaging (a term that combines “inflammation” and “aging”) is a chronic, low-grade inflammation that develops with age. Inflammaging of the immune system is a typical event in HIV-1 infection [126]. The relationship between aging and inflammation is complex and bidirectional [127]. A chronic, low-grade inflammatory state develops as individuals age. This persistent inflammation contributes to aging and the onset of age-related diseases [127]. Many of the alterations affecting the immune system in HIV-1-infected individuals resemble the progression of inflammation in the elderly [126]. HIV-1-mediated inflammation could result from different insults, such as incomplete viral proteins or particles released from the reservoirs, pro-inflammatory cytokines secreted by the cells, HIV-1-increased intestinal permeability, altered gut microbiota, and co-infections [128]. The accumulation of unrepaired DNA damage due to DDR inefficiency, the replicative pressure to compensate for CD4+ T cell depletion, and telomere attrition lead to T-cell exhaustion and senescence, similar to that observed in the elderly [127]. Furthermore, it has been observed that cells with persistent DDRs may trigger the secretion of inflammatory cytokines associated with senescence that may induce a senescent state in neighboring uninfected cells [128]. Although cART effectively controls viral replication, residual inflammaging may persist even in individuals with normal CD4+ T cell counts (but often with an altered T cell CD4+/T cell CD8+ ratio) [129]. Such inflammaging is characterized by short telomeres leading to CD4+ T cell depletion and senescence, similar to those observed in the elderly [110,130].

## 4. HIV-1 Integration and CD4+ T Cell Clonal Expansion

Although at the beginning of an HIV-1 infection, the viruses have identical genomes, these can diversify over time due to the appearance of spontaneous mutations [71,131]. However, clusters of identical viral sequences originating from clonally expanded infected cells are often observed in patients undergoing long-term antiretroviral therapy [132,133]. In patients treated with an efficient ART, plasma viremia can be reduced by more than ten thousand times, while the level of intracellular viral DNA decreases much less, on average, by only fifteen times [134]. This phenomenon has been related to a clonal expansion of infected cells hosting latently integrated provirus [134]. Upon the suspension of ART, there is a rapid increase in plasma viremia. This indicates the presence of some cells in the latent reservoir capable of re-expanding and activating HIV-1 transcription [135,136,137]. Evidence that the provirus often integrates into a cancer-associated gene has suggested that this type of integration potentially can confer a proliferative advantage to clonal populations of infected cells [11]. The viral promoter LTR could grant a proliferative advantage, activating oncogenic gene expression [11]. However, other authors have demonstrated that integrating HIV-1 in cancer-related genes is a minor factor in the clonal expansion of infected CD4+ T cells [138]. Furthermore, HIV-1 proviruses integrated into or close to cancer-related genes are often defective and unable to determine the increase in viremia upon the interruption of ART [138].

Collora et al. found that HIV-1 predominantly resides in granzyme B (GZMB)-expressing cytotoxic effector memory CD4+ T cells. These cells are characterized by their ability to kill infected cells and are marked by the expression of GZMB. The infected cells were often found in large, stable clones that persisted over time, even under ART. Despite ART, persistent antigen stimulation and tumor necrosis factor (TNF) responses were observed. These factors contributed to the expansion of T-cell clones, suggesting that ongoing immune activation drives the proliferation of HIV-1-infected cells. Some HIV-1-infected cells expressed SERPINB9, a protein that may protect them from being killed by CD8+ T cells. Additionally, the expression of BCL2 was noted, which is associated with cell survival. These mechanisms may help infected cells evade the immune system and persist in the body and expand in response to chronic antigen stimulation, allowing the virus to survive long-term [139].

Other authors have observed that the HIV-1 provirus is frequently integrated into a specific category of genes, the KRAB-ZNF genes (KRAB: Krüppel-associated box; ZNF: zinc finger), particularly in patients undergoing long-term ART [12,83,84,85,140,141,142]. KRAB-ZNF proteins produced by these genes are the most abundant family of epigenetic repressors found only in tetrapod vertebrates. Upon binding to DNA, KRAB-ZNF proteins trigger transcriptional repression via interaction with TRIM28, which acts as a scaffold for heterochromatin protein 1 (HP1), H3K9me3-specific histone methyltransferase 1 (SETDB1), and the NuRD histone deacetylase-containing complex. Together, these proteins silence transcription by triggering H3K9 trimethylation and heterochromatin formation [143,144]. Huang et al. analyzed the integration sites of near full-length HIV-1 genomes from individuals on long-term ART. They observed that, in clonally expanded cells of these individuals, proviruses were preferentially integrated within *KRAB-ZNF* genes. Unlike oncogenic genes, *KRAB-ZNF* genes do not promote cell proliferation, so the clonal expansion of CD4+ T cells containing provirus integrations in *KRAB-ZNF* genes can be explained by the lack of production of the virus that is latently integrated and, thus, the lack of cytolytic effect. *KRAB-ZNF* genes are associated with heterochromatin in memory CD4+ T cells, so they are under-expressed [85]. Other studies indicate that HIV-1 proviruses integrated into *KRAB-ZNF* genes are intact and not defective and are only partially refractory to induction, suggesting that these genes might represent the region of host DNA where the provirus can integrate latently but can also be reactivated, for example, by ART withdrawal [12,145].

## 5. Capsid and Its Role in Driving HIV-1 Integration into Gene-Dense Regions or in Lamina-Associated Domains (LADs)

Although HIV-1 integration is primarily mediated by the interaction between integrase and LEDGF/p75 [55], recent studies suggest that capsid interactions with the host factors CPSF6, Nup358, and Nup153 are critical for the selective targeting of integration sites [46,47,146,147]. Experimental depletion of these host factors reduces HIV-1 integration in gene-dense regions [148,149,150]. The primary capsid-binding host factor involved in HIV-1 integration targeting is CPSF6, and CPSF6 knockdown dramatically reduces the targeting of HIV-1 integration in gene-dense regions [39,151]. Tandem knockout experiments of CPSF6 and LEDGF/p75 have shown that CPSF6 plays a more dominant role than LEDGF/p75 in directing HIV-1 integration into gene-dense regions [39]. Furthermore, knockout or knockdown of CPSF6 significantly increases the preference for HIV-1 provirus integration into LADs [151]. LADs are regions of the genome that physically interact with the nuclear lamina. They are characterized by a repressive chromatin environment, low gene density, and limited transcriptional activity. LADs are rich in di- and trimethylated histone H3 lysine 9 (H3K9me2 and H3K9me3)—epigenetic modifications of the DNA-packaging protein histone H3 that promote heterochromatin formation. As a result, genes within LADs are prevalently repressed [152]. LADs are also enriched in GAGA repeats, which serve as binding sites for KRAB-ZNF proteins [153].

Interactions between the HIV-1 capsid and the host factors CPSF6 and Nup153 can be inhibited by capsid inhibitors, an emerging class of anti-HIV-1 drugs [154]. Since CPSF6 regulates integration site selectivity, the treatment of HIV-1 infected cells with capsid inhibitors can influence the sites of HIV-1 integration. Bester et al. demonstrated that the capsid inhibitor GS-6207 (Lenacapavir) reduces HIV-1 integration in gene-dense regions while conversely enhancing integration within lamina-associated domains (LADs) [154]. This pattern parallels the effects observed following CPSF6 knockout or knockdown (Figure 4) [151].

A possible explanation for this phenomenon may be inferred from the work of Emerson et al., which investigated the Gmr1-like family of Gypsy/Ty3-like retrotransposons in the ancestor of amniotes [155]. Many C2H2–type ZFN proteins have a SCAN (SRE-ZBP, CtBP-Associated, and Novel Zinc finger) domain. This is a conserved motif of approximately 80 amino acids found at the N termini of ZFN proteins, involved in protein–protein interactions and capable of dimerization, leading to the creation of homo- and heterodimer protein complexes [156]. SCAN domains are often found alongside KRAB domains [157]. Evidence indicates that the SCAN domain was derived from the C-terminal portion of the gag capsid protein from the Gmr1-like family of Gypsy/Ty3-like retrotransposons in the ancestor of amniotes and was exploited or adaptively coopted for a novel function by C2H2-type ZFN genes [155]. The SCAN domain mediates capsid protein multimerization to form the core retroviral and retrotransposon capsid structure. In addition, protein structural similarity between the SCAN domain and the HIV C-terminal capsid has been observed [155]. There is evidence to suggest that the SCAN domain originally functioned to target ZFN proteins to retroelement capsids [155]. This interaction could influence viral assembly, replication, or restriction. A possible parallel is TRIM5α, a host protein that binds retroviral capsids and prevents infection; although TRIM5α utilizes a RING and SPRY domain, a similar mechanism could apply to ZNF-SCAN proteins [158]. The potential interaction between the capsid and ZFN proteins suggests that capsids unbound to CPSF6 and Nup153 may preferentially interact with ZNF proteins, which are more prevalent in LADs than in chromatin-rich active genes. This interaction may facilitate proviral integration into LADs. In contrast, when the capsid is bound to the host cell factors CPSF6 and Nup 153, the HIV-1 pre-integration complex is directed towards gene-dense regions. Although ZNF proteins with SCAN domains could potentially interact with retroviral capsid elements, direct experimental evidence is limited [155], and this remains an area of potential research in host–virus interactions.

Several outcomes are possible if a retrovirus inserts its DNA into a LAD. Because LADs are less transcriptionally active, an integrated provirus in these regions is more likely to be transcriptionally silent. As a result, the repressive chromatin environment of LADs can help maintain the provirus in a silent state. Insertional mutagenesis considerations can also be made. Retroviral integration in gene-rich and actively transcribed regions can sometimes activate proto-oncogenes or disrupt tumor suppressor genes. By contrast, LADs tend to be less gene-dense. Therefore, integration into LADs might be less likely to cause such overt pathogenic effects. Another question arises: can the proviruses integrated into the LAD form a latent reservoir capable of reactivating upon ART withdrawal? Long-term ART-treated individuals are characterized by large clones of intact proviruses preferentially integrated into heterochromatic regions [142]. Notably, in these individuals, integration is enriched in *KRAB-ZNF* genes, which appear to play a key role in maintaining viral latency and mediating reactivation upon ART interruption [12,145]. Dragoni et al. reported that cells with proviruses integrated into *ZNF* genes (*ZNF470* and *ZNF721*), in locations previously associated with deeper latency, proliferated extensively and produced virus upon stimulation with cognate Gag peptides [12]. Therefore, provirus integrated into *ZNF* genes represents a latent reservoir that can be reactivated. LADs are rich in di- and trimethylated histone 9 lysine and are enriched in GAGA repeats constituting binding sites for ZNF proteins, two conditions markedly repressing gene expression. Currently, no data indicate whether the reactivation of proviruses integrated into LADs is possible. However, it is important to note that LADs are infrequent targets for HIV-1 integration [146]. In contrast, in elite controllers, a significant enrichment of intact proviral sequences in LADs has been observed when clonal intact provirus sequences were counted as independent proviruses [159]. On the basis of these data, we propose a theoretical model in which long-term treatment with capsid inhibitors—over several years or even decades—could lead to the clonal expansion of CD4+ T cells harboring proviruses integrated into LADs and silenced therein (Figure 5). Although this model remains entirely hypothetical at present, it could imply a greater resistance to viral rebound upon discontinuation of capsid inhibitor therapy. Testing this hypothesis would naturally require many years of treatment with this class of anti-HIV-1 drugs.

## 6. Comorbidities Caused by HIV-1-Induced DNA Damage

### 6.1. Cancer and Provirus Integration

The integration of HIV-1 provirus can alter the host genome, potentially contributing to tumor development. Several retroviruses can induce tumors in animals by inserting in or near oncogenes [160]. Retroviral provirus integration can cause DNA breaks or genomic rearrangements, increasing the risk of oncogenic mutations. If the provirus integrates near an oncogene, it can activate it through transcriptional promotion. Moreover, integration can disrupt the function of tumor suppressor genes, facilitating malignant transformation [161]. People with HIV-1 have a higher risk of developing certain types of cancers, known as AIDS-defining cancers (ADCs) and non-AIDS-defining cancers (NADCs) [162]. The pathogenesis of these malignancies is related to the HIV-1-induced immunodeficiency, which allows specific viruses that cause human cancers, such as Epstein–Barr virus, Kaposi sarcoma herpesvirus, and human papillomavirus, to replicate. 

It is well-known that HIV-1 provirus frequently inserts near or into oncogenes. However, such insertions generally do not lead to neoplastic transformation, but they could support the clonal expansion of CD4+ T cells [11]. Moreover, if neoplastic transformation were induced by the insertion of the HIV-1 provirus into or near oncogenic genes, it would be expected to result in T-cell neoplasms. Although lymphomas in people affected by HIV-1 infection are mostly B-cell type, an increased risk of developing T-cell lymphomas has been reported [163]. An increased risk was observed for all subtypes of T-cell lymphoma, including mycosis fungoides, peripheral lymphomas, cutaneous lymphomas, and adult T-cell leukemia/lymphoma (ATLL) [163]. The first case of AIDS-associated non-Hodgkin’s lymphoma in which HIV-1 infection may have played a central role in the lymphocyte transformation process was described by Herndier et al. in 1992. They observed monoclonal-integrated HIV-1 within the genome of T-cell lymphoma, and these cells produced the HIV-1 p24 antigen [164]. In 1994, Shiramizu et al. described four lymphomas in HIV-1-infected patients consisting of large immunoblastic cells. Two were T-cell lymphomas, one was a B-cell lymphoma, and another was described as a Ki-1 positive lymphoma (CD3O) admixed with CD14/CD68-reactive macrophages/histiocytes. HIV-1 p24 was detected in cells phenotypically similar to macrophages in cases 1, 3, and 4 and detected in T cells in case 2. Monoclonal HIV-1 integration sites were observed in all four tumors. The provirus integration site was within the *fur* gene, upstream from the *c-fes/fps* protooncogene [165]. A rare case of B-cell lymphoma in which a defective HIV-1 was integrated upstream of the first *STAT3* coding exon was described in 2007. The provirus had only a 3′ LTR sequence with more vigorous promoter activity than the *STAT3* promoter, and lymphoma cells showed high levels of STAT3, suggesting that the up-regulation of STAT3 caused by HIV-1 integration can induce B-cell lymphoma [166]. STAT3 is implicated in B-cell proliferation and terminal differentiation but can also play an essential role in numerous human malignancies [167]. The few cases of T-cell lymphomas described in the literature, in which the insertion of the HIV-1 provirus has played a role in their pathogenesis, may be attributed to the requirement for additional non-viral mutations to induce human T-cell lymphomas. This is likely the primary reason why proviral insertion in or near an oncogenic gene leads to the clonal expansion of non-neoplastic cells and only very rarely results in a T-cell lymphoma [168]. Some evidence suggests that, in certain individuals with HIV-1 infection, proviral activation of STAT3—and in some cases, both STAT3 and LCK—may compensate for one or more steps required for the development of T-cell lymphomas. HIV-1-infected individuals with proviral activation of these two genes may subsequently acquire additional relevant mutations, potentially leading to the emergence of T-cell lymphomas [168].

### 6.2. Pulmonary Arterial Hypertension

Cardiopulmonary diseases such as pulmonary arterial hypertension have increased significantly in individuals who are HIV-1 infected despite the introduction of ART. Currently, the prevalence of pulmonary arterial hypertension in people with HIV-1 is 2000 times higher than in the general population [15]. HIV-1 pulmonary arterial hypertension is characterized by endothelial hyperplasia and hypertrophy of medial smooth muscle cells [169]. Cells of these tissues lack specific receptors and coreceptors for HIV-1, suggesting that indirect mechanisms allow HIV-1 to induce damage to pulmonary arterial vascular structures. Several reports indicate that two proteins of the virus, Tat and Nef, may play a key role in the pathogenesis of pulmonary hypertension by inhibiting apoptosis induced by components of the DNA repair machinery [15]. Tat is a regulatory protein that enhances the efficiency of viral transcription [170]. It can be secreted from infected cells into the extracellular environment and internalized by uninfected bystander cells [171]. After internalization into the cytoplasm, Tat moves to the nucleus, modulating cellular transcriptional and post-transcriptional processes. Tat interacts with the Tip60 protein, which is involved in DNA damage repair, by inhibiting its activity and facilitating proteasomal degradation [172,173,174]. The suppression of Tip60 blocks the activation of ATM kinase activity and prevents the ATM-dependent phosphorylation of p53 and CHK2. The inhibition and degradation of Tip60 inhibits ATM activity, resulting in the non-repair of DNA and inhibition of p53-induced apoptosis [175]. Nef is an HIV-1 accessory protein that allows immune evasion and the establishment of viral persistence [176]. Nef can be transferred from infected CD4+ T cells into coronary endothelial cells and detected in the endothelial layer of vascular lesions associated with the development of pulmonary arterial hypertension [177]. Nef can activate AKT through a PI3K-dependent mechanism, which induces anti-apoptotic and pro-proliferative signals through Bad phosphorylation/activation [178].

### 6.3. DNA Damage in Renal Tubule Epithelial Cells

HIV-1-associated nephropathy is characterized by focal glomerulosclerosis that typically occurs in the advanced stages of HIV-1 infection, especially in persons of African ancestry [179]. HIV-1-associated nephropathy is characterized by severe proteinuria and rapid progression to end-stage renal disease. In patients treated with ART, the progression to severe renal disease is much slower. HIV-1 can infect renal tubular epithelial cells, podocytes, and parietal epithelial cells, causing cell cycle dysregulation, inflammation, and cell death [180]. Because renal epithelial cells do not express HIV-1 receptors and coreceptors [181], the mechanisms by which renal cells are infected are likely different from those by which HIV-1 infects CD4+ T cells. Ray et al. observed that HIV-1-infected mononuclear cells could transfer the virus to renal tubular epithelial cells by direct cell-to-cell transfer [182]. Other authors have shown that this transfer was partly mediated by heparan sulfate proteoglycans [183]. It was also observed that cultured podocytes from children with HIV-1-associated nephropathy sustained low-level productive infection when exposed to cell-free virus [184]. The ability of cell-free HIV-1 to infect these podocytes depended on the presence of the HIV-1 *env* gene and cell surface proteoglycans [184].

Although the pathogenesis of most renal diseases is still unclear, DNA damage and the related DDR have been observed in acute kidney injury and chronic kidney disease [16]. HIV-1 Vpr protein can induce DNA damage and block DNA replication, the activation of ATR (ataxia telangiectasia and Rad3-related protein), the ubiquitination of histone H2B, and the phosphorylation of histone H2AX that causes chromatin remodeling and the recruitment of DNA repair proteins [185,186]. In addition, Vpr can induce DSBs [7,18,99] and alter cellular transcription through the modulation of NF-kB [7,18,99]. Rosenstiel et al. evaluated the levels of phosphorylated H2AX to determine whether activation of the DNA damage response pathway is involved in HIV-1-associated nephropathy [187]. These authors transduced HK2 cells, a cell line phenotypically similar to primary renal tubular epithelial cells, with lentivirus expressing either HIV-1 Vpr and a *GFP* reporter gene via an IRES intermediate or the GFP reporter alone. Four days after transduction, Vpr-expressing renal tubular epithelial cells were hypertrophic with enlarged nuclei and showed increased phosphorylated H2AX immunofluorescence compared to control cells. Similar results were also observed in in vivo experiments in mice and human renal biopsies from HIV-1-infected patients, where an increase in phosphorylated H2AX was also observed in tubular cells.

### 6.4. Neurological Dysfunctions and DNA Damage

HIV-1 can infect the nervous system, causing central and peripheral nervous disorders. Moreover, the nervous system is an HIV-1 sanctuary in which infections can persist [188]. About 10–20% of HIV-1-infected patients receiving ART are affected by HIV-1-associated neurocognitive disorders that are characterized by impaired memory, anxiety, apathy, mania, and motor dysfunctions [189]. Infection of the nervous system by HIV-1 depends on the ability of macrophages and lymphocytes to cross the blood–brain barrier and infect perivascular macrophages, microglia, and astrocytes, causing damage in neurons, oligodendrocytes, brain microvascular endothelial cells, astrocytes, and microglial cells. Several HIV-1 proteins, including gp120, Tat, Nef and Vpr, can directly induce neuronal cell death [190]. Numerous scientific reports indicate that impaired DNA repair mechanisms and the accumulation of DNA damage in nervous system cells can lead to various neurological syndromes [17,191,192,193,194]. Consequently, DNA damage is likely a pathogenic factor in the development of central and peripheral nervous system dysfunction, while the proper functioning of DNA repair mechanisms may play a crucial role in preventing pathological conditions. Increasing evidence suggests that Vpr contributes to neuropathogenesis in individuals infected with HIV-1 [195]. The neurotoxic effects of Vpr have been attributed to various factors, including alterations in cell cycle progression, the dysregulation of cellular metabolism, and disrupted signaling [195]. However, given both the role of Vpr in inducing DNA damage and the growing evidence that DNA damage plays a significant role in the pathogenesis of numerous neurological syndromes, it is plausible to suggest that Vpr-induced DNA damage may be a central factor in the nervous system disorders observed in HIV-1-infected patients. Moreover, Vpr’s ability to inhibit DNA repair mechanisms could further contribute to the accumulation of DNA damage in nervous system cells. Interesting data have emerged from studying DNA damage caused by stabilizing DNA–topoisomerase complexes associated with defects in DNA repair systems. Topoisomerases are enzymes that catalyze changes in the topological state of DNA, interconverting relaxed and supercoiled forms through transient breaks of single-stranded (topoisomerase 1) or double-stranded (topoisomerase 2) DNA. During this activity, a cleavage intermediate complex is formed from the end of the DNA covalently bound with the tyrosine of the topoisomerase active site. Under certain conditions, this complex can be stabilized, preventing DNA binding and triggering a DDR. Neurological dysfunctions caused by the accumulation of topoisomerase-mediated DNA damage have been observed, leading to the conclusion that topoisomerase activity may play a relevant role in these pathologies [196,197]. There are several similarities between Vpr-induced DNA damage and the mechanisms of nervous system dysfunction caused by the stabilization of topoisomerase–DNA complexes. As reported by Iijima et al. [8], the Vpr-induced unwinding of DNA causes the accumulation of supercoiling of DNA and the formation of topoisomerase 1–DNA covalent complexes. Therefore, Vpr may be involved in the development of HIV-1-associated neurocognitive disorders by the accumulation of supercoiling of DNA and the formation of topoisomerase 1–DNA covalent complexes that cause DSBs [8]. The inhibition of DNA repair systems by Vpr would prevent the repair of these DSBs, making the brain dysfunction model in HIV-1 very similar to other non-HIV-1-related models described in the scientific literature [198,199,200].

## 7. New Therapeutic Strategies to Treat HIV-1 Infection

### 7.1. HIV-1 Treatment Based on Capsid Inhibitors

Currently, most ART regimens use one or two drugs belonging to the NRTI class associated with an integrase or protease inhibitor. NRTIs can induce significant side effects, including lactic acidosis, pancreatitis, and renal failure [201,202,203]. Furthermore, several works have shown that NRTIs cause telomere shortening, resulting in cellular senescence and inflammation [118,119,120,121,122,123,124,125]. Currently, integrase inhibitors are the first-choice drugs for the treatment of HIV-1 but are often combined with NRTIs. Capsid inhibitors are an emerging class of anti-HIV-1 drugs. Several key HIV-1 capsid inhibitors are either approved or under investigation. GS-6207 (Lenacapavir) was approved in 2022 by the FDA for treatment-experienced individuals with multidrug-resistant HIV-1 [204]. It binds to a conserved arginine-rich region within the capsid hexamer pocket at the interface of CA monomers, inhibiting capsid disassembly (uncoating) and assembly in a dose-dependent manner. This mechanism affects both the early (nuclear import, reverse transcription) and late (assembly, maturation) stages of HIV-1 replication. The binding of Lenacapavir allosterically alters the conformation of the capsid, disrupting the recruitment of host factors, such as CPSF6 and Nup153. CPSF6 functions in nuclear import and directs PIC to the host cell genome’s nuclear speckles and gene-rich regions [28]. Without the capsid–CPSF6 interaction, PIC is precluded from reaching nuclear speckles and gene-rich regions of chromatin [205]. Current clinical trials are investigating Lenacapavir as an oral or subcutaneous agent as part of a dual therapy for both already treated and untreated patients. In phase 2 studies, the combination of Lenacapavir and Bictegravir treatment was well tolerated and induced a rapid reduction in plasma viremia in HIV-1-infected patients and was highly effective in maintaining virologic suppression [206,207,208,209]. As reported in Section 5, the LEDGF/integrase and CPSF6/capsid complexes have different effects on the integration sites of HIV-1; moreover, Lenacapavir inhibits provirus integration into active genes, promoting integration into LADs, a condition that may favor the silencing of integrated proviruses. Future studies will be needed to understand the therapeutic potential of this drug combination in long-term-treated patients. GS-CA1 and PF74 are capsid inhibitors currently in preclinical or early clinical trials [208]. They share a similar mechanism of action to GS-6207; however, PF74 binds near the host factor interaction sites, directly competing with the host factors CPSF6 and Nup153 [210].

### 7.2. The “Shock and Kill” Therapeutic Approach

The “Shock and Kill” strategy is an experimental approach aimed at curing HIV-1 by targeting and eliminating latent viral reservoirs that persist despite ART. This method involves two primary steps: (i) shock (latency reversal), utilizing latency-reversing agents to reactivate dormant HIV-1 within infected cells, making the virus detectable; and (ii) kill (elimination), where reactivated infected cells are targeted and destroyed by the immune system or therapeutic interventions. Several clinical trials have been conducted to evaluate the efficacy of this strategy. The ROADMAP study assessed the combination of the latency reversal agent romidepsin with the broadly neutralizing antibody 3BNC117. Despite reactivating latent HIV-1, the approach did not significantly reduce the viral reservoir [211]. Vorinostat, an HDAC inhibitor, was studied for its potential to reverse HIV-1 latency. While it successfully reactivated latent HIV-1 in some studies, eliminating the reactivated cells remains a challenge [212]. Studies have explored combining latency-reversing agents with agents that promote the apoptosis of infected cells, such as BCL-2 inhibitors. However, an optimal combination of a latency-reversing agent and a BCL-2 inhibitor that enables the maximal reactivation and selective elimination of HIV-1 has yet to be identified [213].

### 7.3. HIV-1 Treatment Based on Silencing or Knocking out the Provirus

Advancements in understanding the epigenetic regulation of HIV-1 have led to the development of novel therapeutic strategies, currently in the experimental stage, for treating individuals infected with the virus. The “Block and Lock” approach is an experimental strategy for achieving a functional cure for HIV-1, aiming to silence the virus permanently rather than eliminate it. In contrast to the “Shock and Kill” method, which seeks to reactivate and eradicate latent HIV-1, “Block and Lock” focuses on maintaining the virus in a deeply latent state, preventing its reactivation even in the absence of antiretroviral therapy [214]. To date, the only successful example of the “Block and Lock” strategy is the use of didehydro-cortistatin A, a specific inhibitor of the Tat protein, which can achieve the long-term suppression of HIV-1 reactivation through the establishment of epigenetic silencing [215,216,217]. Didehydro-cortistatin A inhibits HIV-1 by specific transcriptional and DNA methylation changes in genes related to the cell cycle, histones, and interferon responses. While preclinical studies have yielded promising results, translating this strategy into clinical practice remains challenging. Ensuring the long-term safety and efficacy of latency-promoting agents is crucial. Ongoing research continues to refine these approaches and evaluate their potential in human clinical trials. CRISPR-based gene editing is a promising strategy for curing HIV-1 infection by directly targeting and removing the viral genome from infected cells. CRISPR-Cas systems, particularly CRISPR-Cas9, cut and edit the HIV-1 genome or host genes involved in viral infection. This system consists of three main strategies: (i) HIV-1 provirus elimination (“Cut and Remove”), in which CRISPR is programmed to target and excise the HIV-1 provirus from the DNA of infected cells [218]; (ii) blocking HIV-1 entry (“Gene Knockout”) through gene-editing approaches aims to mimic mutations that make cells resistant to HIV-1 infection [219]; and (iii) disrupting HIV-1 replication (“Gene Disruption”) by CRISPR that disables essential HIV-1 genes (e.g., tat, rev, gag), preventing the virus from replicating [220]. This approach can keep latent HIV-1 from reactivating, reducing viral reservoirs. A prominent strategy involves the experimental therapy EBT-101, developed by Excision BioTherapeutics, Inc. EBT-101 utilizes the CRISPR-Cas9 system to excise HIV-1 proviral DNA from infected cells. Preclinical studies demonstrated its potential to remove HIV-1 DNA from various cell lines and animal models. In 2021, the U.S. Food and Drug Administration (FDA) authorized initiating clinical trials for EBT-101. Later, in July 2023, the FDA granted EBT-101 Fast Track Designation, accelerating its development process. Although preclinical animal studies have demonstrated promising curative potential, recent data presented at the 27th ASGCT meeting in Baltimore revealed that EBT-101 failed to prevent viral recurrence in three participants who discontinued antiretroviral therapy during the phase 1/2 trial, requiring them to restart ART treatment [221].

## 8. Conclusions

The intricate interactions between HIV-1 and the host genome, explored in this review, offer valuable insights into the underlying reasons for the poor outcomes in achieving a definitive cure for HIV-1 infection despite the use of the most advanced therapeutic strategies. Factors such as DNA damage induced by HIV-1, the inhibition of DNA repair pathways, telomere attrition, the orientation of the provirus within host genes, clonal expansion of CD4+ T cells harboring the provirus in a latent form, and integration into specific genes such as oncogenes or *KRAB-ZNF* genes are likely among the key contributors that hinder progress toward a lasting cure. A more in-depth study of cases where HIV-1-infected individuals have achieved a prolonged aviremic state without ART could guide research toward more effective therapeutic strategies. Bone marrow transplantation (BMT) has shown potential in eradicating HIV-1 in some instances, although it remains an exceptionally rare occurrence. The most notable examples are the cases of Timothy Ray Brown, known as the “Berlin Patient”, and the “London Patient”, both of whom received stem cell transplants from donors carrying the CCR5-Δ32 genetic mutation, which confers resistance to HIV-1 infection [222,223]. These cases suggest that a gene-editing approach to mimic this mutation, rendering cells resistant to HIV-1 infection, could represent a feasible therapeutic strategy. The study of mechanisms through which elite controllers maintain aviremia or low viremia without ART could guide research towards more effective strategies for treating HIV-1 infection [224]. The aviremic or low-viremia state in elite controllers is often attributed to a combination of host genetic, epigenetic, and viral factors, such as the CCR5-Δ32 mutation, highly efficient HIV-1-specific CD8+ T cells, enhanced natural killer cell activity, increased production of interferons and other antiviral cytokines, and HIV-1 proviruses subject to epigenetic silencing. However, replicating this unique interaction of factors as therapeutic strategies in HIV-1-infected patients by combining treatments such as “Cut and Remove”, “Gene Knockout”, and “Gene Disruption” remains a challenge. Lenacapavir stands out as an innovative drug due to its unique mechanism of action, distinguishing it from NRTIs, NNRTIs, protease inhibitors, and integrase inhibitors. Its use in combination with other antiviral drugs, such as Bictegravir or Islatravir [225], appears both safe and effective. Furthermore, its long-acting properties have the potential to significantly improve the quality of life for HIV-1-infected patients. Future studies will be needed to understand the therapeutic potential of capsid inhibitors in combination with other classes of anti-HIV-1 drugs in long-term-treated patients.

## Figures and Tables

**Figure 1 ijms-26-03184-f001:**
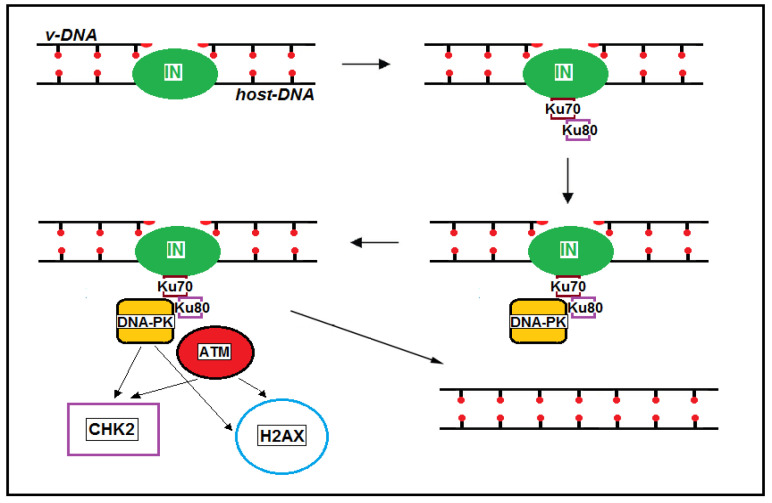
The viral integrase and Ku70 complex recruits Ku80, DNA-PK, and ATM at the integration site. ATM phosphorylates CHK2 and H2AX, two key proteins that initiate activation of the DNA damage checkpoint. CHK2 inhibits CDC25, preventing cell entry into the S and M phases. The phosphorylated form of H2AX is a scaffold that attracts repair factors to DNA damage sites and retains them until the damage is removed.

**Figure 2 ijms-26-03184-f002:**
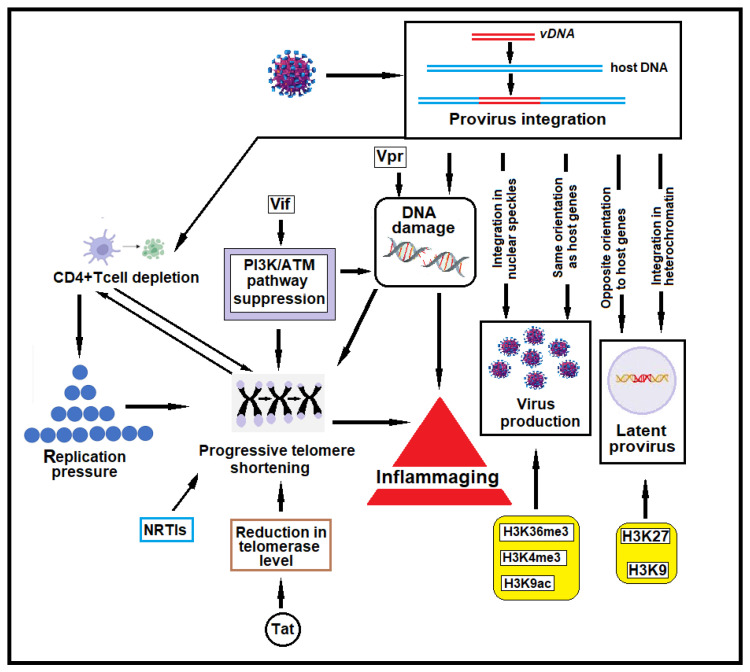
Schematic representation of the complex interactions between HIV-1 and the host cell genome. The factors that promote virus production are as follows: (i) integration into nuclear speckles; (ii) integration in the same orientation as host genes; and (iii) integration within genes marked by H3K36me3, H3K9ac, or H3K4me3 epigenetic histone modifications. Integration into heterochromatin, in the opposite orientation to host genes, or in genes marked by H3K27 or H3K9 histone epigenetic modifications promotes viral latency. Integration and Vpr can cause DNA damage, which is not repaired due to PI3K/ATM suppression by Vif. Replication pressure caused by HIV-1-induced CD4+ T cell depletion, PI3K/ATM suppression, DNA damage, reduced telomerase levels, and treatment with NRTIs leads to progressive telomere shortening. Telomere shortening is a cause of CD4+ T cell depletion and, together with DNA damage, contributes to inflammaging.

**Figure 3 ijms-26-03184-f003:**
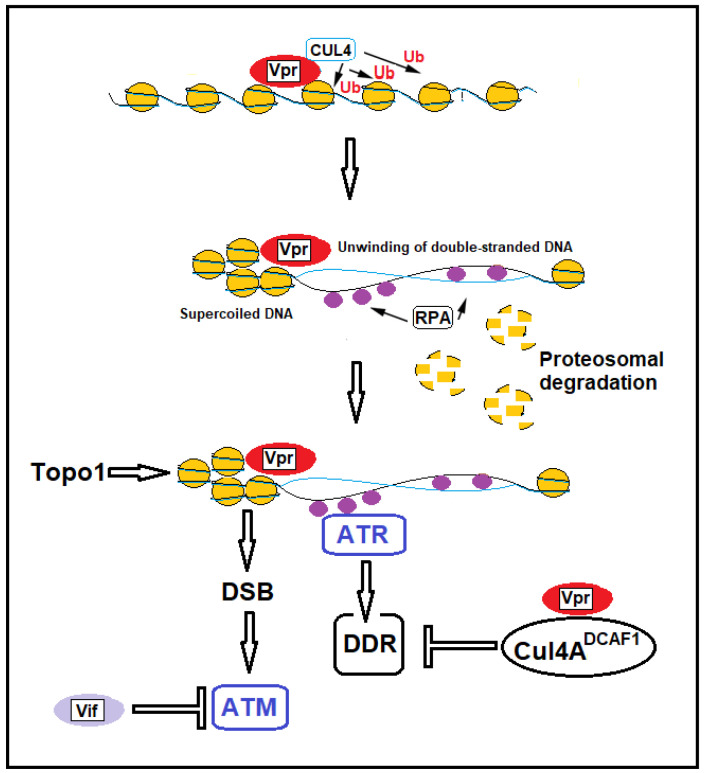
Vpr unwinds double-stranded DNA by converting it into a relaxed form and induces the ubiquitination of histone H2B. This chromatin remodeling promotes the efficient loading of RPA70 that activates the ATR-dependent DDR. In addition, the Vpr-induced unwinding of double-stranded DNA results in the accumulation of negatively supercoiled DNA and covalent complexes of topoisomerase 1 and DNA, which cause DSBs. Vpr promotes the degradation of several DDR proteins, inhibiting double-strand DNA break repair by recruiting the CRL4ADCAF1 ubiquitin ligase complex. Vif blocks the Vpr-directed activation of ATM but not ATR.

**Figure 4 ijms-26-03184-f004:**
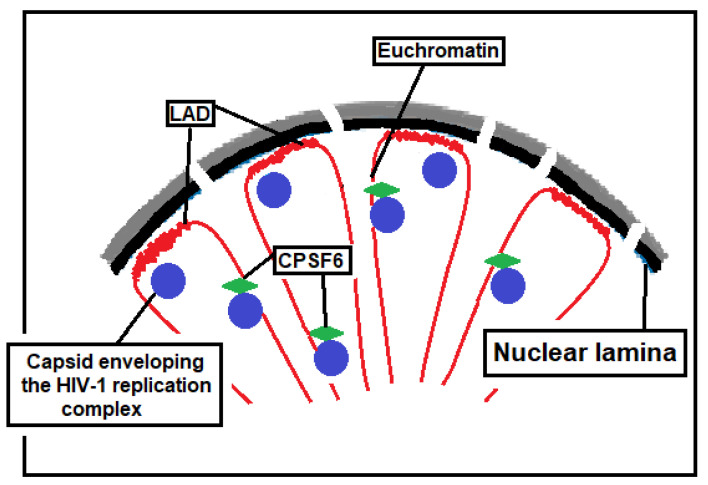
CPSF6 plays a predominant role in directing HIV-1 integration into gene-dense regions of euchromatin. Knockout or knockdown of the host factor CPSF6 or treatment with Lenacapavir results in a significant preference for HIV-1 provirus integration into LADs. The repressive chromatin environment of LADs may help maintain the provirus in a silent state.

**Figure 5 ijms-26-03184-f005:**
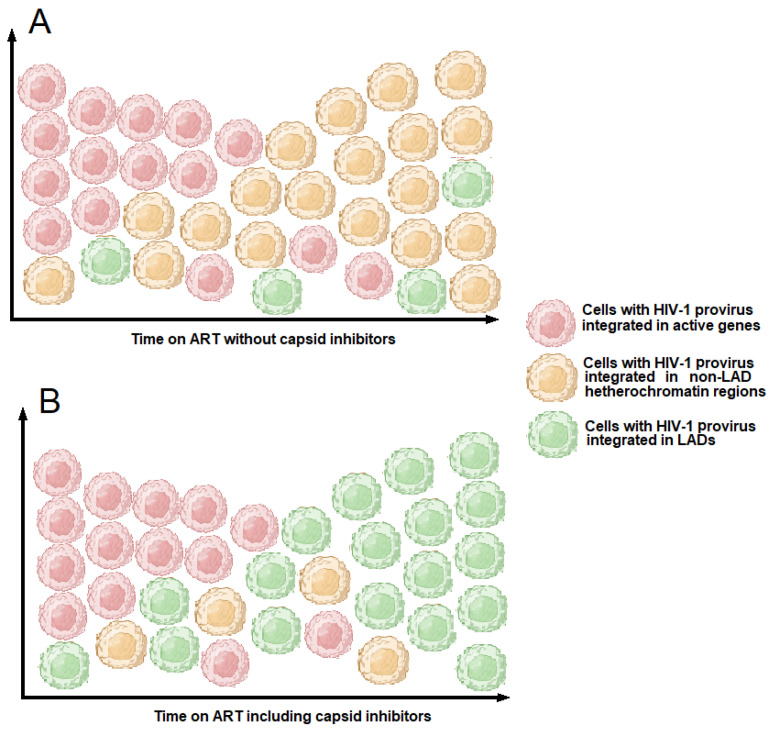
(**A**) Long-term ART-treated individuals are characterized by large clones of intact proviruses preferentially integrated in heterochromatin locations. (**B**) A theoretical model in which long-term treatment with ART including capsid inhibitors could lead to the clonal expansion of CD4+ T cells harboring proviruses integrated into LADs and silenced therein.

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
