# Peer review of "The Complex Interactions Between HIV-1 and Human Host Cell Genome: From Molecular Mechanisms to Clinical Practice"

_ijms, 2025, doi:10.3390/ijms26073184_

Round 1
Reviewer 1 Report
Comments and Suggestions for Authors
Tolomeo and coll have undertaken a detailed review on both basic, therapeutic and translational aspect dealing with HIV infection.The review is updated and well organized. Just a few observations:
1)Lines 199-200 in pag 5 : “ a selection of intact proviruses with viral latency characteristics” page 5 line 199-200 could be changed in “ intact latent HIV integrated “occurs …
2) Fig 2: since the form of the manuscript is a review please change figure 2 in more clear graphic representation including representation of T cells, telomers etc. In the present for is too schematic.
3) Lines 358-361 page 9: “During DNA replication, the enzyme DNA polymerase cannot fully replicate the ends of linear DNA, leading to the progressive shortening of chromosomes” .explain why.
Author Response
Comments 1: Lines 199-200 in pag 5 : “ a selection of intact proviruses with viral latency characteristics” could be changed in “ intact latent HIV integrated “occurs …
Response 1: We have changed the sentence “a selection of intact proviruses with viral latency characteristics” in “ intact latent HIV integrated “.
Comments 2: Fig 2: Since the form of the manuscript is a review please change figure 2 in more clear graphic representation including representation of T cells, telomers etc. In the present for is too schematic.
Response 2: We have revised Figure 2 with clearer graphical representations. However, due to its complexity and space constraints, we have included only the most essential elements to enhance clarity.
Comments 3: Lines 358-361 page 9: “During DNA replication, the enzyme DNA polymerase cannot fully replicate the ends of linear DNA, leading to the progressive shortening of chromosomes”. Explain why.
Response 3: In the revised manuscript we have explained in detail why the enzyme DNA polymerase cannot fully replicate the ends of linear DNA, leading to the progressive shortening of chromosomes.
Reviewer 2 Report
Comments and Suggestions for Authors
In the review entitled 'The Complex Interactions Between HIV-1 and Human Host Cell Genome: From Molecular Mechanisms to Clinical Practice', Manlio et al focused on HIV integration and DNA damage caused by HIV and HIV treatments.
This is a good review and provided comprehensive opinion. There are a few places can be improved:
1. Under section 3.2 and 3.3, the authors discussed HIV-induced telomere shortening. And under section 7.1, the authors mentioned telomere shortening can also be caused by some ART regimens. While the telomere shortening is one of the key discussion points in the manuscript, the clarification of how telomere shorted should be described more precisely and maybe consider putting them into the same section?
2. The correlation from HIV-induced gene profile difference and cancer should be more precise: The cancer type should be introduced as each cancer has its own markers. (e.g. Line 456, BACH2 is more a cell proliferation marker than a cancer marker)
3. In T cell clonal expansion part, I recommend the authors review papers related to GZMB and the HIV-specific TCR clones.
4. There are still several typos alongside the manuscript. Here are some examples: Line 295&296&299, NF-κB; Line 346, HIV-1;
Author Response
Comments 1: Under section 3.2 and 3.3, the authors discussed HIV-induced telomere shortening. And under section 7.1, the authors mentioned telomere shortening can also be caused by some ART regimens. While the telomere shortening is one of the key discussion points in the manuscript, the clarification of how telomere shorted should be described more precisely and maybe consider putting them into the same section?
Response 1: In the revised manuscript, in section 3.2 we have elucidated how telomeres shorten explaining in detail why the enzyme DNA polymerase cannot fully replicate the ends of linear DNA, leading to the progressive shortening of chromosomes.
Comments 2: The correlation from HIV-induced gene profile difference and cancer should be more precise: The cancer type should be introduced as each cancer has its own markers. (e.g. Line 456, BACH2 is more a cell proliferation marker than a cancer marker).
Response 2: In the revised manuscript we specified that BACH2 is mainly involved in cell proliferation, reporting the specific reference. However, we also reported a work were BACH2 was identified as a novel partner gene in IGH translocation in a patient with highly aggressive B-cell lymphoma/leukaemia, and another work demonstrating that the insertion of pro-viruses at the BACH2 locus in a murine model of B-lymphomagenesis provides strong evidence that mutations at this locus contribute to disease development. In addition we specified the types of cancer correlated to genes involved in HIV-1 integration.
Comments 3: In T cell clonal expansion part, I recommend the authors review papers related to GZMB and the HIV-specific TCR clones.
Response 3: We added in T cell clonal expansion part of revised manuscript a section in which we discuss the role of HIV-specific TCR clones expressing GZMB.
Comments 4: There are still several typos alongside the manuscript. Here are some examples: Line 295&296&299, NF-κB; Line 346, HIV-1.
Response 3: Typos alongside the manuscript have been corrected.